:ᐧᵂᵂ: PLOS | ONE

# An experimental erythrocyte rigidity index (Ri) and its correlations with Transcranial Doppler velocities (TAMMV), Gosling Pulsatility Index PI, hematocrit, hemoglobin concentration and red cell distribution width (RDW)

**Antonio Valadão Cardoso**◯ *

Rheology Laboratory, Materials Engineering Post-Graduation Program REDEMAT-UEMG/DESP-ED, State University of Minas Gerais, Belo Horizonte, Minas Gerais, Brasil

* antonio.cardoso@uemg.br

## Abstract

Brain artery velocities (Time-Averaged Maximum Mean Velocity, TAMMV) by Transcranial Doppler (TCD), hematocrit, hemoglobin, Red blood cell (RBC) Distribution Width (RDW) and RBC rigidity index (Ri), when reported together with their correlations, provide a accurate and useful diagnostic picture than blood viscosity measurements alone. Additionally, our study included a sixth parameter provided by TCD, the Gosling Pulsatility Index PI, which is an indicator of CBF (Cerebral Blood Flow) resistance. All these parameters are routine in Hematology except for values of Ri. The rigidity (Ri) of the RBC is the main rheological characteristic of the blood of Sickle Cell Anemia (SCA) patients and several pathologies. However, its quantification depends on many commercial and experimental techniques, none disseminated and predominant around the World. The difference in absorbance values of the blood, during the process of sedimentation in a microwell of a Microplate Reader, is a straightforward way of semi-quantifying the RBC rigidity Ri, since the fraction of irreversibly sickled red blood cells does not form rouleaux. Erythrocyte Rigidity Index (Ri) was calculated using initial absorbance $A_{initial}$ (6 s) and final $A_{final}$ (540 s), $Ri = 1 / (Ai-Af)$. The Ri of 119 patients (2–17 y / o, M & F) SCA, SCC (Sickle Cell/hemoglobin C), SCD (Sickle Cell/hemoglobin D), Sβ0thal (Sickle Cell/hemoglobin Beta Zero Thalassemia) and 71 blood donors (20–65 y / o, M & F) were measured in our laboratory while the five parameters (TAMMV and PI by TCD, Hct, Hb and RDW) were obtained from medical records. The *in vitro* addition of hydroxyurea (HU, 50mg /dl, n = 51 patients, and n = 8 healthy donors) in the samples decreased the rouleaux adhesion strength of both donor and patients' blood samples, leading to extraordinarily high Ri values. The correlation between the studied parameters was especially significant for the direct relationships between Ri, TAMMV, and PI.

**Data Availability Statement:** All relevant data are within the paper and its Supporting Information files.

**Funding:** Sources of funding for this study: Cemig GT S/A (Centrais Elétricas de Minas Gerais). There was no additional external funding received for this study. The funders had no role in study design, data collection and analysis, decision to publish, or preparation of the manuscript.

**Competing interests:** Financial support for the project was carried out with Brazilian public money because Brazilian law requires companies in the electricity sector to invest in R&D through public notices open to all researchers. I received a non-refundable scholarship during the project. In the subject of this article, I have no commercial obligation to CEMIG whatsoever, including patents, employment, consultancy, products in development, marketed products, etc. This does not alter my adherence to PLOS ONE policies on sharing data and materials.

## Introduction

More than 50 years ago, researchers who decisively helped implant the field of hemorheology —such as L. Dientenfass, S. Chien, H. Schmid-Schonbein, and R. Wells—described blood as an emulsion. The rheological behavior of healthy blood at 37 $^{o}$ C is understood as liquid drops (hemoglobin suspension) lined by a lipid membrane inside another liquid (the plasma). Only then would it be possible to know how a hematocrit Hct = 95 (0.95 l / l) - 100 [1,2] flows. In addition, other studies [3,4] have shown that at 37°C, deformable red blood cells decrease blood viscosity as a result of a rotation of the RBC membrane around cell content, which makes cell characteristics comparable to those of a drop of fluid. In a human erythrocyte, the absence of cytoskeletal elements in the cytoplasm allows the erythrocyte membrane to rotate around the cytoplasm, leading to a tank tread-like motion [5].

In sickle cell anemia (SCA), the deoxygenated RBC cytoplasm stops flowing (or flows very little) due to the polymerization of deoxyhemoglobin S. Hemoglobin S (HbS) is a mutant of normal hemoglobin HbA (A, adult). HbS results from the replacement of valine by glutamic acid at the sixth position of the beta-globin chain. Inheritance of one sickle globin gene leads to sickle cell trait, while the inheritance of two sickle globin genes results in sickle cell anemia. The cause of sickle cell erythrocyte gelation was evident by electron microscopy imaging showing nanometer-diameter HbS fibers [6]. The polymerization of hemoglobin in the form of fibers and the occurrence of sickle cell is a fact reported since the twentieth century in several animal species [7]. Although the causes of polymerization are unclear, it should be borne in mind that the transport of blood $O_2$ through vertebrates is optimized by high erythrocyte hemoglobin (Hb) concentrations near the solubility limit [8]. And recent findings suggest that symmetrical protein complexes, such as hemoglobin, exist on the edge of supramolecular self-assembly [9].

Hundreds of studies [10] and especially numerous new techniques have been proposed to quantify RBC aggregation and deformability, which helps to understand the origin of symptoms of various diseases, including SCA. The quantification of RBC deformability has received increasing attention from researchers [11,12,13,14] and various experimental techniques (filtration, micropipette, ektacytometry, Dientenfass equation based on viscosity and hematocrit values [15,16], etc.) that analyze the deformability—or its inverse, the stiffness of the RBC— have been proposed and applied over the last 50 years [17,18,19,20,21,22,23].

Recent studies that visualize blood flow in the brain (4D MRI) tend to relativize the relevance of SCA blood viscosity in stroke [24,25]. Although the blood velocity in the major cerebral arteries in SCA children is much higher than in healthy children and adults, the shear rate and shear stress in the endothelium of the Circles of Willis in SCA patients would be *lower* than in the control group [26,27,28]. Generally, they support the view that may be the viscosity, at high shear rates (arterial circulation), *is not* a predictor for stroke processes in SCA patients [29,30,31,32]. However, nothing has been said about the rigidity of the red sickle cell [33] as a predictor.

Tests with SCA suspensions and normal RBCs in microfluidic devices [34,35,36,37] suggests that:

A–RBC rigidity/deformability is a function of the internal viscosity of hemoglobin suspension, of RBC membrane and volumetric alterations

B—the presence of HbF (fetal hemoglobin, main oxygen transport protein in the human fetus and newborns) reduces this internal viscosity as it reduces or avoids polymerization

C—rigidity is the leading cause of the tendency of RBC to travel towards the walls while deformable RBCs travel mainly in the center area of the vessels [38,39,40].

In clinical studies, Bowers [41] has observed lower deformability (higher stiffness) in the RBC of SCA patients with leg ulcers (LU). However, blood viscosity would not be determinant in leg ulcers (LU), because SCA patients with LU and without LU present the same values at low shear rates (microcirculation) [42].

Unlike what is common to hear among physicians, the blood viscosity of SCA is lower than that of healthy people because they have Hct almost half of normal, and the hematocrit is the determinant of the viscosity value.

The technique considered the gold standard [43] for measurement of RBC is laser ektacytometry. Although progress has been made in evaluating a population of heterogeneous (deformable and non-deformable) erythrocytes [44,45] the presence of rigid red blood cells interferes with the results, and this fact recognized since the introduction of the technique [46]. Developed decades ago [47] very few laser ektacytometry pieces of equipment do exist at hemocenters of middle- and low-income countries but with a high incidence of SCA patients or related hemoglobinopathies.

Syllectograms was the term coined by W. G. Zijlstra [48] in 1957 to describe the transmittance/reflectance/ absorbance of light passing through a blood sample in the process of rouleaux formation. Sillectograms can be obtained on various types of equipment. In the present work, a simple technique (difference between the initial and final values of the absorbance A) recorded using a microplate reader, was utilized to measure, semi-quantitatively, the RBC rigidity (Ri) of blood samples from patients SCA, SCC, Sβ0thal, and SD and compare them with Ri values of healthy adult donors. The objective is to verify if the overall rigidity of red blood cells is increased in these infant patients (2–17 years).

An essential advantage of the technical principle proposed here is that no substance (apart EDTA anticoagulant) needs to be added to the blood sample; it is possible to obtain the rigidity index Ri directly from 100 microliters of the patient's blood sample. Another advantage is that the microplate reader is a common technique even in hemocenters in low- and middle-income countries. This type of evaluation of the rigidity of the RBC may help, along with the blood count data, for the clinical assessment in places where more sophisticated equipment is not available such as laser ektacytometry equipment.

Transcranial Doppler ultrasound (TCD) is a noninvasive technique that evaluates velocity, direction, and other properties of blood flow in the basal cerebral arteries through a pulsed ultrasonic beam. Flow velocities are consistent with direct invasive flow measurements [49]. Additional information on cerebral hemodynamics may derive from the TCD waveform, and the most commonly used is the Gosling Pulsatility Index PI [50], which describes the pulsatility of TCD waveforms. PI is the difference between systolic and diastolic flow velocities divided by the mean velocity. PI is considered a reflection of the distal cerebral vascular resistance (CVR).

Possibly for the first time, we seek to correlate these TCD parameters with the RBC stiffness. The diagram of Fig 1 summarizes the most relevant investigated quantities in the analysis of human blood rheology and its relevance to CBF:

A- Hematocrit is determinant in CBF (Cerebral Blood Flow) values with inverse correlation with hematocrit [51,52] and also a determinant for the viscosity value;

B- Velocities in the main cerebral arteries (MCA: middle cerebral artery, ACA: anterior cerebral artery, PCA: posterior cerebral artery, ICA: internal carotid, BA: basal artery) by TCD are a noninvasive measure for blood velocity;

C- Gosling's pulsatility index PI is a hemodynamic parameter provided by TCD and considered an indicator of mechanical resistance to CBF [53,54];

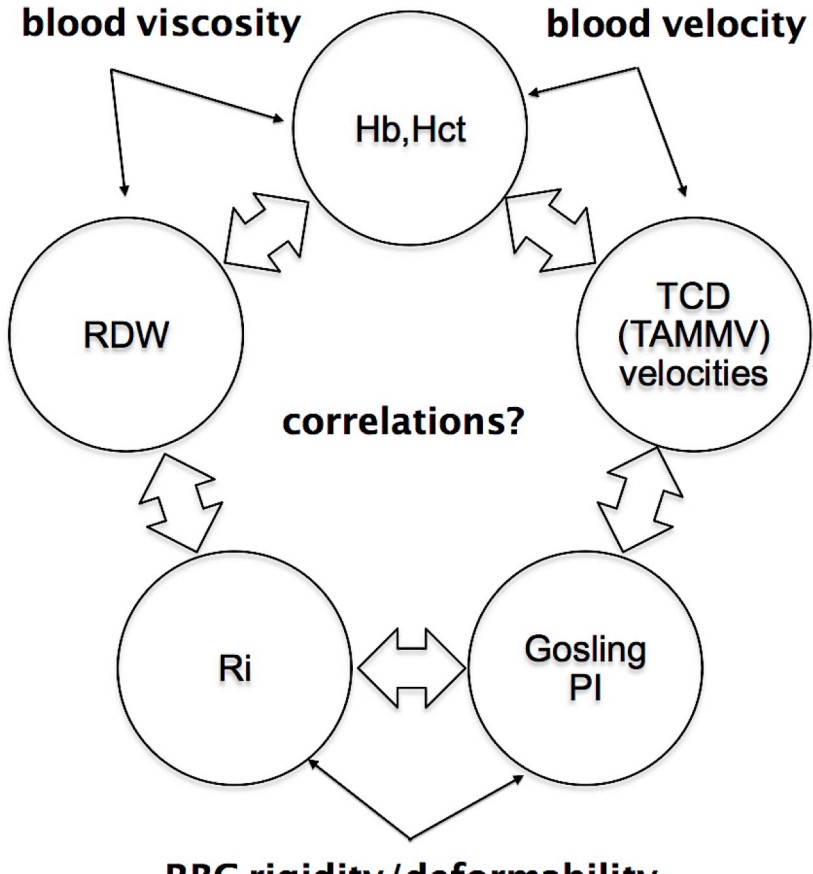

**Fig 1. Investigating correlations to improve understanding of flow in cerebral arteries properties by correlating it to relevant hemodynamic parameters TCD TAMMV velocities, Hct, RDW, Gosling PI, and a proposed RBC rigidity index Ri.**

D- Ri is the RBC rigidity index proposed in this work and influences blood rheology;

E- Red Cell Distribution Width (RDW) is a measure that affects CBF because it is related to the size/volume dispersion distribution of RBC.

Correlations between these values may provide a more accurate picture of the individual's hemorheology rather than solely the blood viscosity value. These parameters are usually already supplied in the blood count, and the Ri rigidity index obtained without purchasing expensive equipment.

The objective of the work is to investigate these correlations shown in Fig 1.

## Materials and methods

### 2.1- Subjects

The study involved one hundred and sixty nine people: 71 donors and 98 patients (68 HbSS patients, 21 HbSC patients, 6 HbSBzerothal, and 3 HbSD). Patients regularly attended by the Hemominas Foundation (acronym HBH), a governmental blood center in Belo Horizonte, Minas Gerais, Brazil. All the patients were children of both sexes, ages between 2 and 17 years

in steady-state health conditions; and adult donors aged between 20–65 years. Both the legal representatives of the children and the donors have signed a consent form to participate in the study. Two projects were approved by the Ethics Committee (CEP) of the HBH (Records Nos. 190 and 327) and conducted in accordance with the recommendations of the 1964 Declaration of Helsinki and reviews.

## 2.2- Collection and treatment of samples

**2.2.1**- **Donors.** Donors were recruited after medical screening from people who had already donated blood to HBH. Sample collection for the project was performed prior to sample collection for serological and immunohematological tests. Vacuum collection tubes (5ml) containing EDTA as an anticoagulant were used. All donors were over 18 years old.

**2.2.2**- **Patients.** 56 boys, 42 girls being 68 SCA, 21 SCC, 6 Sβ0thal and 3 SCD-trait. 13 patients were on transfusion therapy (Tx) and hydroxyurea (HU), 32 were on HU therapy only and 53 who did not use HU or Tx; hematocrit (Hct) in the range of 16 to 36%. Patients accompanied by parents or legal guardians were approached when they arrived for blood collection for laboratory tests at the HBH outpatient clinic. Sample collection for this project was performed at the same time as venipuncture for other medical examinations. The inclusion criteria of the patients were: a) having SS, SC, SD or Sβ0thal hemoglobinopathies; b) be under 18 years of age. In the collection were used vacuum tubes (5 ml) containing EDTA as an anticoagulant.

**2.2.3**- **Venipuncture was performed between 7:00 and 12:00 in the morning and blood samples were analyzed within 4 hours to avoid rheological changes in the blood [55].** After collection, the sample tubes were placed upright on the polystyrene support to prevent direct contact with ice and stored in a cool box containing ice at the bottom and then transported to the Rheology Laboratory. Upon arrival at the laboratory, the samples were stored in a homogenizer at 10˚ C refrigeration until analysis began and after complete blood oxygenation. Guidelines for international standardization in blood rheology techniques and measurements were strictly followed [56,57].

## 2.3 Rigidity of red blood cells (RBC)

**2.3.1**- **Selection of hematocrit to perform the tests.** The technique described in this article has been developed in our laboratory in recent years, initially with animal blood (pig and bovine) and then with human blood [58,59,60,61,62]. The hematocrit was selected after testing on various hematocrit through different studies (with human, swine and bovine blood samples) using the developed method, the Hct hematocrit = 1,5,10,20,30,40,50 and 60 were tested. In addition, the evaluation of substances such as fibrinogen and albumin [62,63], added in vitro, has also been tested in the past.

The RBC Rigidity Index Ri test was safely performed on hematocrit> 10%. The option for Hct = 20 was defined to accompany previous studies with this hematocrit value. In addition to the adjusted hematocrit assays, whole blood (Hct = native) assays were performed on various patient blood samples. For Hct = native, the whole blood sample was injected into the microwell without any treatment, i.e., as it was collected with the added EDTA. The objective of performing whole blood and adjusted hematocrit tests was to verify if there was a significant difference in the rigidity index Ri values between the two samples.

**2.3.2**- **Sample preparation.** Hematocrit (Hct) was obtained by microcentrifugation (model 211, FANEM). Hematocrit adjustment was performed by blood centrifugation (5000RPM, 10˚C for 5 minutes, Hettich MIKRO 220R), plasma removal and addition of cells in the desired ratio to obtain 20% (v / v). For assays with Hct = native, no hematocrit

adjustement was made. The samples were aliquoted into labeled 1.5 ml microtubes (Eppendorf), homogenized under refrigeration at 10˚ C until testing. This procedure was performed in air to ensure the oxygen saturation of the sample [64].

**2.3.3- Absorbance measurements.** A microplate reader (model 680, BIORAD) was used for absorbance measurements (a.u.). An aliquot of the sample was taken from the homogenizer and used in a container to facilitate pipetting and oxygenation. The parameters for the tests were: test temperature at 37˚ C, wavelength (lambda) of 655nm, and 90 readings were taken at 6 (six) second intervals between measurements. The 655 nm wavelength was selected because it is outside the absorption band of Hb, MetHb (Methemoglobin), OxHb (Oxyhemoglobin), and CarbHb (Carbaminohemoglobin) [65,66] to avoid absorption from free Hb in plasma. 100 µl of the blood sample placed in the U-shaped bottom 96-microwell plates. A single operator performed all assays in this study. First, the equipment operator was trained to eject the microplate in the shortest possible time. The results obtained were automatically recorded by the computer and then transferred to the electronic spreadsheet software (Excel). Some types of micropipettes were tested: single channel, multichannel and positive displacement, all Gilson brand. The single channel pipette (Gilson) was chosen for the study. The use of the technique is described in references [67,68].

**2.3.4 –Contact (wetting) angle measurement (theta) of the microwell [69].** Assays were performed to evaluate contact angle between liquid (blood plasma at room temperature) and the plastic of the microwell of two commercial trademarks of microplates (number 1: GTT and number 2: Labtest). The surface of the microplate number 1 has a wetting angle (θ) of ~ 58˚, and the surface of microplate number 2 has a theta of ~ 100˚ (**See S4 Fig**). Microplate number 1, due to wetting, was used in most tests.

**2.3.5—The following formula calculates the Rigidity index (Ri).**

$$Ri_{RBC} = \frac{1}{(Ainitial - Afinal)} \qquad (1)$$

Where $A_{inicial}$ and $A_{final}$ are the absorbance values (arbitrary units) of the first and last reading in the assay. Two types of Ri measurements were performed, one with Hct adjusted for Hct = 20 ($Ri_{20}$) and one for Hct = native ($Ri_{Native}$). In these two hematocrits, the absorbance values were always well below the detection limit of the microplate reader (abs = 4.0 a.u.). The nomenclature Ri was used to differentiate from the nomenclature of rigidity indexes of RBC measured by other techniques such as micropipette and viscometry.

**2.3.6- Test duration: 540 seconds in total (90 readings).** After 250–300 seconds, the tendency is for the absorbance value (A) to stabilize, indicating that the sedimentation process has ended. Assay proceeds to 90th reading to verify an increase of A, indicative of a reduction in erythrocyte aggregation due to an increase of the absorbing area of the incident beam.

**2.3.7- In vitro addition of hydroxyurea (HU) powder in the samples.** *2.3.7.1—HU in vitro addition.* Having in mind the observations of previous studies [70,71], the HU powder (Hydrea) was removed from capsules and weighed on an analytical balance (model AUX-220, Shimadzu) in the calculated quantity and added to an aliquot of the blood sample. After this sample aliquoted in 1.5ml microtubes (Eppendorf), labeled and homogenized in air and under refrigeration at 10˚ C until the test was carried out.

*2.3.7.2- Sample selection for HU addition.* In 48 patients and seven donor samples, 50 mg /dl hydroxyurea (HU) was added in an aliquot of blood, specially prepared for this test. The HU trial was performed only on patient samples that reported never having used the drug. In the case of donors, the selection of samples for the in vitro addition of HU was randomized.

**2.3.8- Calculation of shear rate applied in the erythrocyte Ri rigidity test.** The shear rate calculation considered the ejection force of the Gilson model P1000 micropipette and D200 tip (brand Gilson, exit diameter: 0.85 mm). The force at the moment of pipetting (data provided by the manufacturer) is in the range of 10-30N and the shear rate (lambda, s-1) at the time of ejection can be higher than 1000 s$^{-1}$ [(sigma (Pa) = F/A and lambda (s-1) = sigma (Pa)/ blood viscosity (Pa.s)]. Lambda (s-1) ~ 1000 s$^{-1}$ is higher than the values utilized on commercial ecktacytometers. The usual range of blood disaggregation shear rate reported is 50–150 s$^{-1}$, ensuring 100 ul was ejected disaggregated.

## 2.4- Analysis of blood samples by optical microscopy

Samples received for both patient and donor assays were investigated by light microscopy (Leica, model DM-LS) at room temperature. Using a micropipette (Gilson), a drop of the blood sample (Hct = 20) was gently deposited on a glass slide and covered with a coverslip, the blood film trapped only by surface tension. The general condition of the sample, the erythrocyte aggregation, and the occurrence of crenated red cells were investigated. Microstructures of 39 samples recorded: 28 from patients (14 males and 14 females) and 11 from donors (5 females and 6 males). Images are of blood with hematocrit set to H = 20. In each of these samples, at least 20 pictures of the blood microstructure recorded at 40X, 200X, 400X and 1000X magnifications.

## 2.5—Hematocrit values (Hct), hemoglobin (Hb) and Red blood cell Distribution Width (RDW)

**2.5.1- Hematocrit (Hct in %), hemoglobin concentration (Hb in g/l), and RDW (%) of patients and donors, were obtained from the HBH (governmental blood center) medical archive.** At HBH, a Coulter T-890 hematology counter was used to perform all blood cell counts.

Data collection in the HBH archives was performed directly in the patients' medical records. In possession of the date of blood collection to measure Ri, we searched the patient's folder for the blood count performed on the date closest to the date of collection for the project. The HBH has not permitted the reproduction by any means of these patient medical records. It only allowed copying the blood count data. Since errors can be made in the process of collecting and transcribing data manually, the coincidence of dates between Ri measurement and blood counts is not assured. On the other hand, donor Hb data are from the same date of collection to measure Ri, as it is a blood donation procedure.

**2.5.2—TCD velocities (TAMMV) in the left and right cerebral arteries MCA, ACA, MCA/ACA, PCA and t-ICA (terminal internal carotid artery) and basal artery (BA) and Gosling Pulsatility Index PI of patients obtained from the medical file of the patients at the archives of HBH (Government Blood Center).** Data collected, when possible, from TCD closest to the RBC rigidity test (Ri). At HBH Transcranial Doppler (TCD), examinations at 2 MHz were performed and interpreted by a single expert using Viasys equipment (model Companyon III, Viasys, USA)

## 2.6- Statistical analysis

Results are presented as means ± SD (standard deviation). All variables were checked for normal distribution with the Kolmogorov-Smirnov test. When comparing two groups, Mann-Whitney U-test, Kolmogorov-Sirmov, and Wald-Wolfowitz simultaneously performed. Kruskal-Willis performed when comparing more than two groups. The correlation analyses were performed between variables using Rank Correlations (Spearman Rho and Pearson r). A two-

sided p-value of <0.05 was considered statistically significant. All statistical analyses performed on Statplus: Mac Pro v6.1.60, AnalystSoft Inc.

## Results

### 3.1. Semi-quantitative rigidity index of the RBC, Ri (Hct = 20) and Ri (Hct = native): Calculation

Fig 2 shows the result of RBC Ri rigidity essays performed in a- sample of a male donor (p <0.001), b-male SS patient (p <0.001) and c-female SC patient (p <0.001), Hct = 20. The value of the RBC rigidity index Ri is much lower for the donors. The difference between the initial and final absorbance values is higher for the donors because the deformable red cells form rouleaux and allow more light to pass (greater transmittance), at the end of the sedimentation process inside the microwell. Furthermore, Fig 2 shows that the slope of the curves in the range of 100 to 200 seconds seems to indicate that the slope is more pronounced for donors compared to unhealthy cells. Further work needs to be done to confirm and clarify the origin of these differences.

Blood samples from donors—at the time of ejection by the micropipette- present initial absorbance (Ainitial) values higher than Afinal since the flexible and deformable red blood cells separate at the ejection moment to immediately re-organize into rouleaux during sedimentation. The initial absorbance of the SS sample reflects the fact that the small SS aggregates,

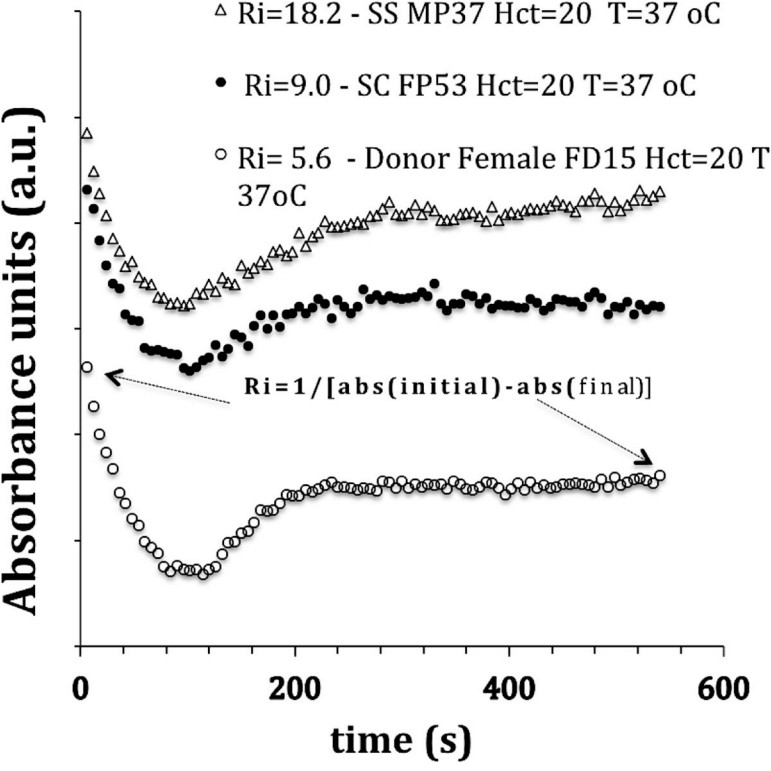

**Fig 2.** Microplate Reader essays (absorbance, a.u.) performed to calculate (semi-quantitative) RBC Ri rigidity index values in a) sample of a male donor (p <0.001), b) SS male patient (p <0.001) and c) SC female patient (p <0.001); Hct = 20 (0.2 l/l) in the all essays. The value of the RBC rigidity index Ri is lower for the donors. The difference between the initial and final absorbance values is higher for the donors because the deformable red cells form rouleaux and allow more light to pass (greater transmittance) at the end of the sedimentation process inside the microwell. When sedimentation finally ends, the absorbance is constant because the rouleaux are stable.

and to a lesser extent SC, requires more stress [72,73] to separate. After 100 seconds, the sedimentation of the 20uL of red blood cells (Hct = 20) practically finished. From this point, the increase in absorbance directly related to the inability of the irreversible sickle cells (ISCs) SS to organize themselves into rouleaux because of their rigidity. The SS blood sample consists of red blood cells that aggregate much less [74,75,76], so the final absorbance (Afinal) of the sample is only slightly less than the initial absorbance (Ainitial). The difference between initial and final absorbance follows the order $deltaA_{donor} > deltaA_{SCC} > deltaA_{SCA}$. As Ri = 1 / delta (Absorbance), the larger the difference, the smaller the Ri or, the lower the overall rigidity of the red blood cells.

Absorbance (abs) presents a dynamic during the test period describing a curve with a minimum value (t ~ 100s) followed by an increase in absorbance up to t ~ 200s. The minimum abs value is related to the end of the 2D rouleaux formation process (like piles of coins). The increase in absorbance after ~100s might be associated with the formation of 3D structures of connected and sedimented rouleaux that, while contracting and compacting, reduce light transmission. In the case of donors, after 200s, we have a line parallel to the abscissa or even a slight reduction in absorbance due to more effective compaction. In the case of SS patient curves, a slight increase may occur (Fig 2) at the end of the trial because the hardened red cells do not form stable structures.

Table 1 presents the hemorheological parameters of patients and donors.

(Fig 3A and 3B) presents a comparison between the values of the Ri rigidity index at different hematocrits for healthy donors and SCA patients. For this donor sample, no significant difference was observed between Ri values, even doubling the Hct (from 20 to 40). Although the curves are different, the delta (A_initial-A_final) is very close. In the case of patient SCA, the values of Ri for Hct = 20 and Hct = native are close but differ from the Ri in Hct = 40. Possibly the concentration of irreversibly polymerized SS cells increases with Hct and affects the value of Ri. Considering the results presented in Fig 3, we can speculate that in order to obtain Ri for healthy donors, one should keep the native hematocrit.

Fig 4 shows the difference between the Ri values of a- SCA patients who have not used HU and neither transfusion (Tx) and donors (p <0.001); b- SCA patients with and without HU and Tx therapies (p <0.02); c- SCA patients using HU and donors (p <0.001) and d-SCC patients and donors (p <0.001). The coefficient of variation (CV) of the rigidity measurements Ri of SCA patients is high, mainly because the experimental procedure used manual techniques, not automated.

Fig 5 presents a comparison between the Ri rigidity index values for SCA and SCC patients in the hematocrit Hct = 20 and Hct = native (blood sample only with EDTA anticoagulant). The Mann-Whitney comparison tests were performed to verify whether the difference between the Ri values of SCA patients at Hct = 20 and Hct = native was significant in each of the conditions presented (patient without HU and TX therapies p < 0.87, with HU therapy but no TX p <0.45, with both HU and TX therapies p <0.64). In neither of these alternatives

**Table 1. Age, gender, and hematological parameters (Hb, Hct, RDW), and Ri (rigidity index) of patients and donors.**

| Hematological parameters | SCA (n = 68) | SCC (n = 21) | Sβ0 Thal (n = 6) | SCD (n = 3) | Controls (n = 71) |
|---|---|---|---|---|---|
| Age (years) | 2–17 | 2–13 | 4–15 | 7–13 | 20–65 |
| Gender (male/female) | 35/30 | 11/10 | 2/3 | 3/0 | 35/36 |
| Haemoglobin Hb(g/dL) | 8.4± 1.1 | 10.7±0.9 | 8.5±1.4 | 10.5± 2.8 | 15.0±1.4 |
| Hematocrit Hct(%) | 25.0±3.6 | 32.8±3.0 | 27.8±4.5 | 31.6±8.2 | - |
| RDW(%) | 19.5±4.4 | 16.5±2.9 | 21.1±4.8 | 15.6±2.3 | - |
| Rigidity Index RI = | 19.7±18.8 | 9.9±4.0 | - | - | 6.7±3.5 |

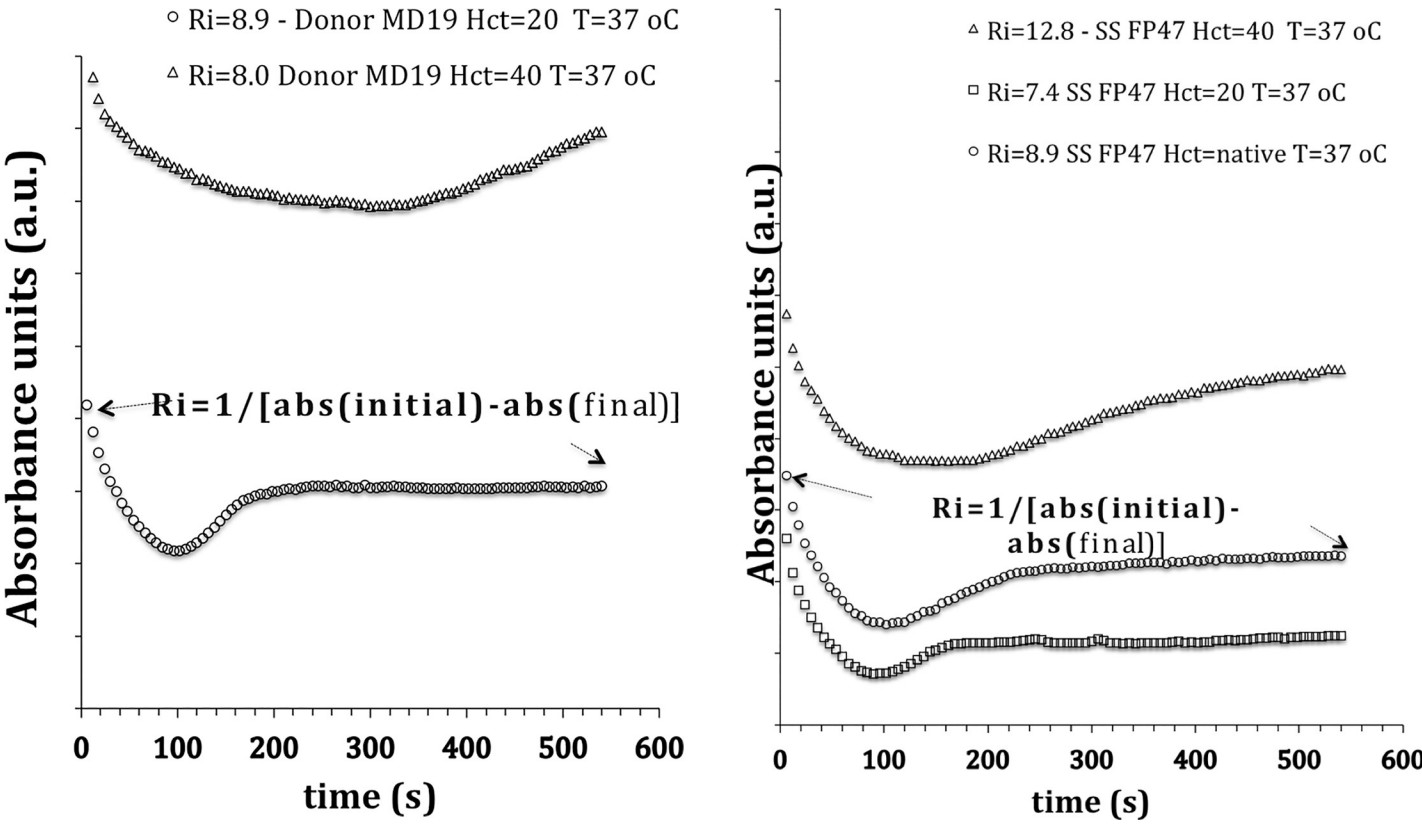

**Fig 3.** (a) Run to semi-quantitatively evaluate the Rigidity indexes of the same healthy donor sample. Ri at Hct = 20 (0.2 l/l) and Hct = 40, (b) Run to semi-quantitatively evaluate the RI rigidity indexes of a SCA patient sample at Hct = 20, Hct = native and Hct = 40. Assays performed with 100 μl of blood at T = 37° C; samples with EDTA anticoagulant only.

was there any significant difference in Ri value in both hematocrits. The lowest Ri values are for SCA patients, and SCC who use both therapies and the largest Ri are for patients who do not use either treatment.

Table 2. A presents the values of the RBC rigidity Ri for samples with Hct = 20 and Hct = native. In the tests with Hct = native, the sample injected *into the well without any alteration or preparation*. The content of plasma and other blood cells did not significantly influence the rigidity of the red blood cells (p <0.29), especially in the case of SCA patients without HU and transfusion therapies.

In Table 2B rigidity (Ri), data obtained for samples with addition in vitro of 50 mg /dl of HU in the blood samples from patients SCA, SCC, and donors in hematocrit adjusted at Hct = 20.

Huang et al. [77] studied the RBC deformability of SS patients and healthy individuals, RBC incubated in vitro with HU, and observed that while HU stiffened SS RBC (possibly by methemoglobin formation), HU did not affect normal RBC. In the present work, this was not observed. There was a significant increase in the Ri rigidity index of RBC with the in vitro addition of HU (c = 50 mg /dl) indistinctly affecting SS RBC and AA RBC as shown in the box-plot of Fig 6. Usually, it is considered that HU is chemically transformed in the liver [78].

In general, in our experiments, absorbance curves of added HU in vitro (c = 50 mg/dl) showed the initial (first measured) absorbance value higher than those without HU both patient and donor samples, indicating that they were more disaggregated during ejection by

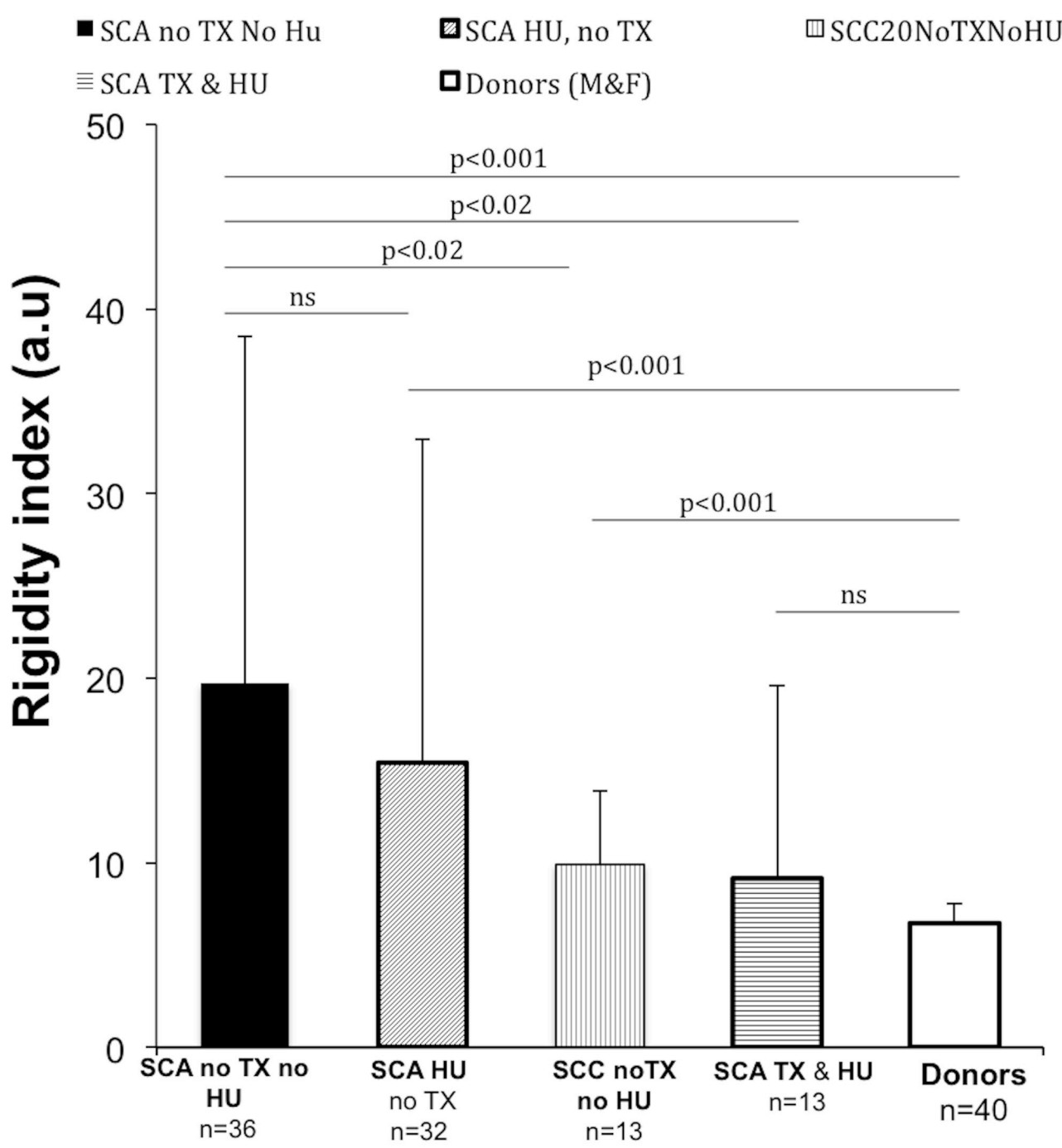

**Fig 4. Values of the rigidity index of red blood cells (Ri).** Differences between donor Ri values and a- SCA patients with (p <0.001) and without transfusion (Tx) and hydroxyurea (HU) therapies (p <0.001) and b- SCC patients (p <0.001) were significant. Also significant is the difference between c- the Ri values of SCA patients with and without transfusion therapy (p <0.02). The difference between Ri values of SCA patients with and without treatment with HU is not significant (ns). Also, no significant difference found between Ri of patients on HU and Tx and Ri values of donors, indicating that transfusion reduces further Ri than only treatment with HU.

the micropipette. However, at the end of the assay, the last measured absorbance value (Afinal. See Eq 1) were virtually equal or had a higher value than the first measured (Ainitial),

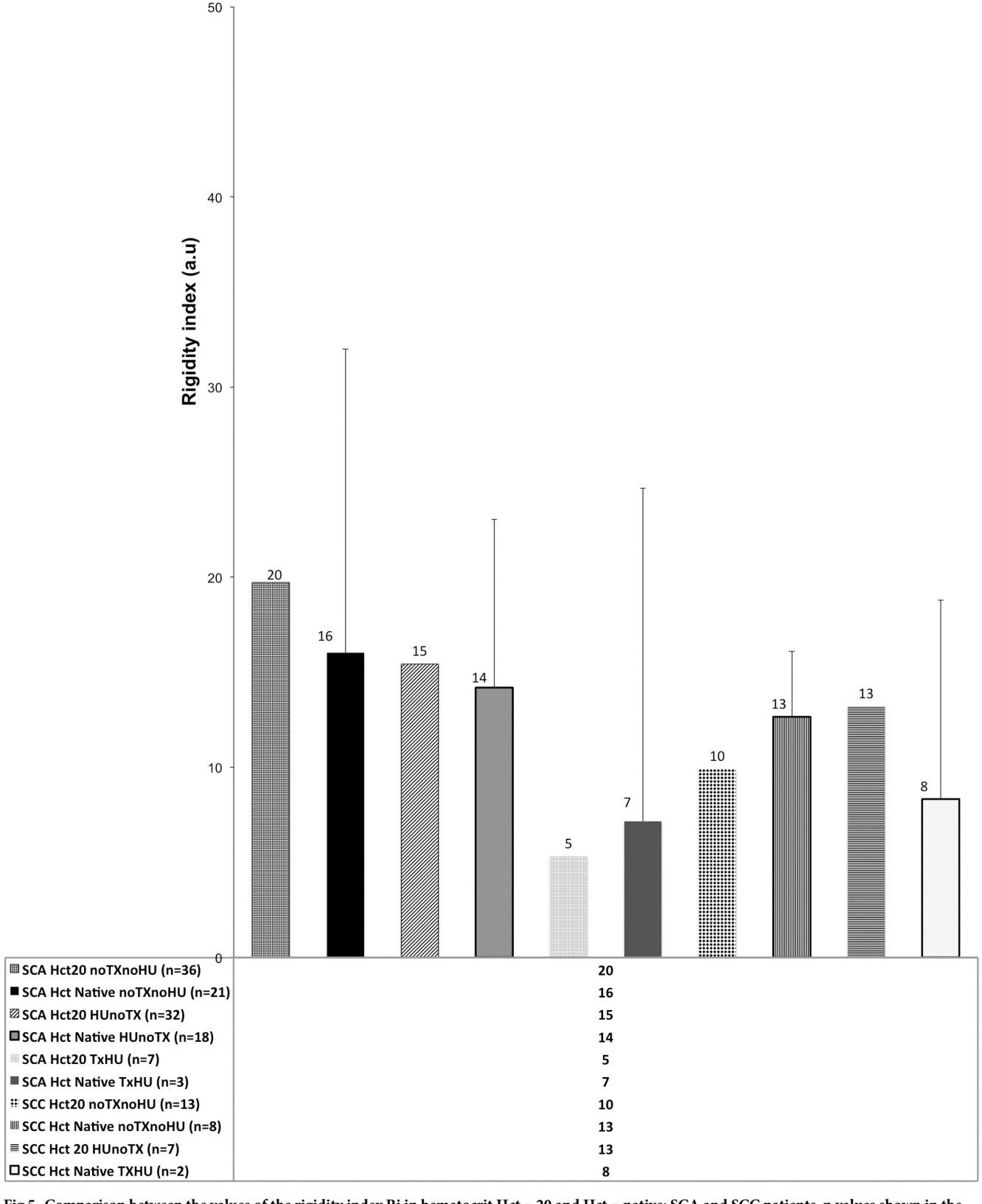

**Fig 5. Comparison between the values of the rigidity index Ri in hematocrit Hct = 20 and Hct = native; SCA and SCC patients, n values shown in the figure.** HU = in hydroxyurea therapy, TX = in transfusion therapy. NoHU = does not use hydroxyurea, no TX = does not use transfusion. Significance between the two data series (Hct = 20 and Hct = native), using the nonparametric Mann-Whitney test.

**Table 2. Rigidity index Ri for Hct = 20 and Hct = native.** 2.A. Ri for patients (2–17 y/o, male and female); 2.B. Ri values for the vitro HU (50mg/dl) added to the blood sample. Values presented as mean ± standard deviation.

| A. Ri Patients (2–17 y/o, M&F), Hct = native and Hct = 20 (0.2 l/l). | | | | | | | | Healthy Donors (M&F) | |
|---|---|---|---|---|---|---|---|---|---|
| Hemoglobinopathies | | Therapy | | | | | | | |
| | | No HU no TX | | HU No TX | | HU & TX | | | |
| SCA | Hct(%) | 20 | native | 20 | native | 20 | native | 20 | native |
| | n | 36 | 21 | 32 | 18 | 7 | 2 | 40 | 3 |
| | Ri (avg+stdev) | 20±18 | 16±16 | 15±18 | 14±9 | 5±4 | 7±0.6 | 7±3 | 8±2 |
| SCC | n | 13 | 8 | 7 | - | - | 2 | | |
| | Ri | 11±3 | 13±3 | 13±7 | - | - | 8±5 | | |

| B. Ri values for the vitro HU (50mg/dl) added to the blood sample. | | | |
|---|---|---|---|
| Patients not under TX or HU therapy Hct = 20 | | Healthy Donors (M&F) Hct = 20 | |
| SCA | n | 19 | 6 |
| | Ri (avg+stdev) | 29±19 | 42±62 |
| **SCC** | **n** | **9** | |
| | **Ri (avg+stdev)** | **55±105** | |
| **Sβ0 thal** | **n** | **4** | |
| | **Ri (avg+stdev)** | **17±8** | |

TX = Transfusion; HU = Hydroxyurea;M&F = Male and Female

indicating that the RBC aggregates were spontaneously separating. Changes in aggregation/ disaggregation possibly due to changes in the blood zeta potential [79], and because HU dissolved in plasma has acted as an electrolyte, increasing the net electrical charge on the surface of the erythrocytes. HU dissolved (50mg/dl) changes the plasma electrolyte balance, and the human blood Zeta potential. Zeta is measured usually around ~ 13mV, which in theory [80] favors aggregation. Zeta changes are, therefore, physiologically critical [81] because they alter RBC surface properties, affecting the rheology of hemoglobin suspensions (AA and SS) and the erythrocyte membrane [82].

Fig 7 shows the assays of one patient and one donor with and without in vitro addition of HU. The addition of HU produces a clear absorbance difference of the sample from ~200 seconds after the start of the assay in both patient and donor. This 3-minute variation cannot be attributed to hemoglobin denaturation because, according to Roa [83], Hb denaturation under HU action takes days. The increase in absorbance cannot be attributed either to the occurrence of hemolysis because the wavelength used in the photometric assays (lambda = 655 nm) is outside the absorption band of hemoglobin, oxyhemoglobin and methemoglobin [84,85].

Fig 8 shows the relationship between Ri values before and after the addition of HU (50 mg /dl). The trend-line indicates a nonlinear correlation (semi-log graph) for Ri growth when HU added in vitro. The same behavior occurs with donor blood; that is, the in vitro addition of HU makes Ri grow extraordinarily. Increased absorbance after red cell sedimentation (~200 seconds after pipetting) indicates that red blood cells are probably separating from aggregates and creating more light absorption area. Therefore, the values obtained in this experiment suggest that the effect of HU was to weaken the adhesion of red blood cells at 37˚C. Ri values much higher than those found for SCA and SCC patients indicate another phenomenon, not related to rigidity and yet not elucidated.

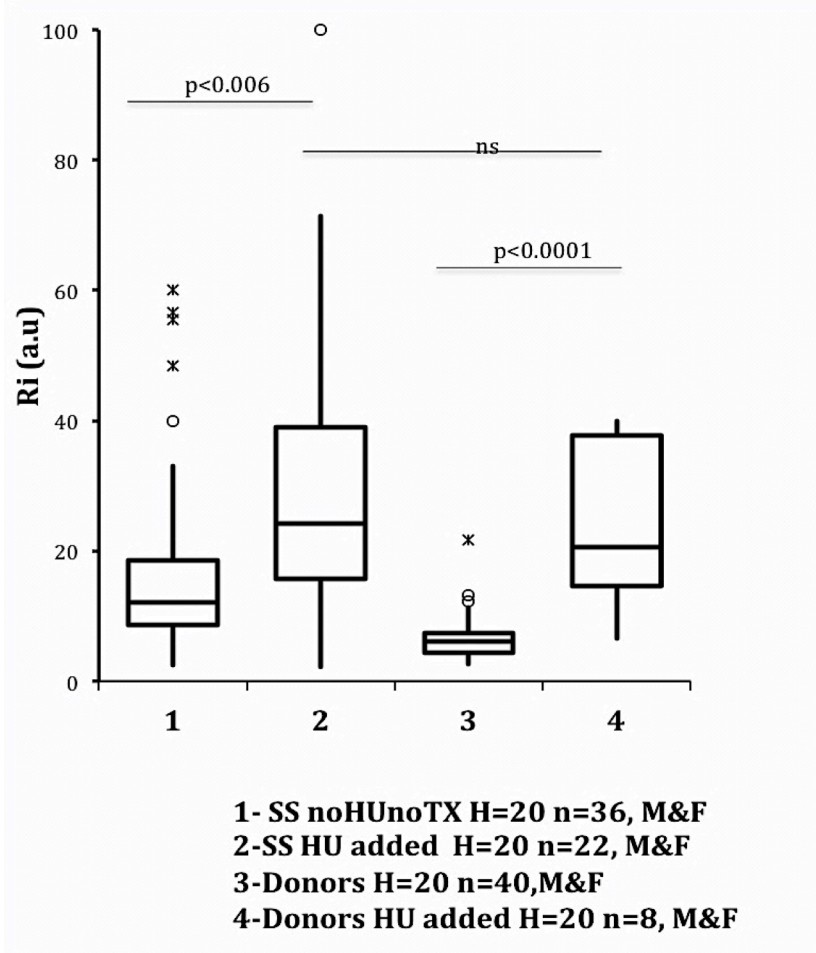

**Fig 6. RBC Ri rigidity index values of SS (n = 36) and donor (n = 40) samples without (c = 0 mg/dl) and with addition in vitro of HU (c = 50mg/ml).** Significant differences in Ri values between SS without/with HU (p <0.006) and donor without/with HU (p <0.001) samples. No significant differences in Ri values were observed between SS and donors with HU. All assays performed at 37˚ C and Hct = 20.

## 3.2. Rigidity index Ri (at Hct = 20 and Hct = native), TCD velocities (TAMM), and PI for patients

Fig 9 presents the correlations between TCD velocities (TAMMV, time-averaged mean maximum velocity in cm/s) of the left and right cerebral arteries (MCA, MCA / ACA, ACA, PCA, t-ICA) and basal artery (BA) data for patients SCA, SCC, SD, Sβ0thal with Ri. Also shown in Fig 9 are the correlations between TAMMV with Hb, Hct, and RDW. All correlations presented in Fig 9 are significant; see **S1 Table** at Supplemental Material for n and P-values. The correlations of Ri with the velocities in MCA (R = 0.42, p <0.001) and ACA (r = 0.39, p <0.001) are weak but significant. More importantly, they are at the same level as the correlations of the parameters Hb, Hct, and RDW. A note of caution is that the n value of Hb, Hct, and RDW are higher than the n of Ri values. As observed in other studies [43], RDW correlates negatively with Hb and Hct. The greater the anemia, the higher the RDW value, as already observed in previous studies [43,86].

Correlations of the Rigidity index (Ri) with the velocities (TAMMV) in all the arteries are positive (see Fig 9). Also, the correlations of TAMMV are positive for RDW. On the other

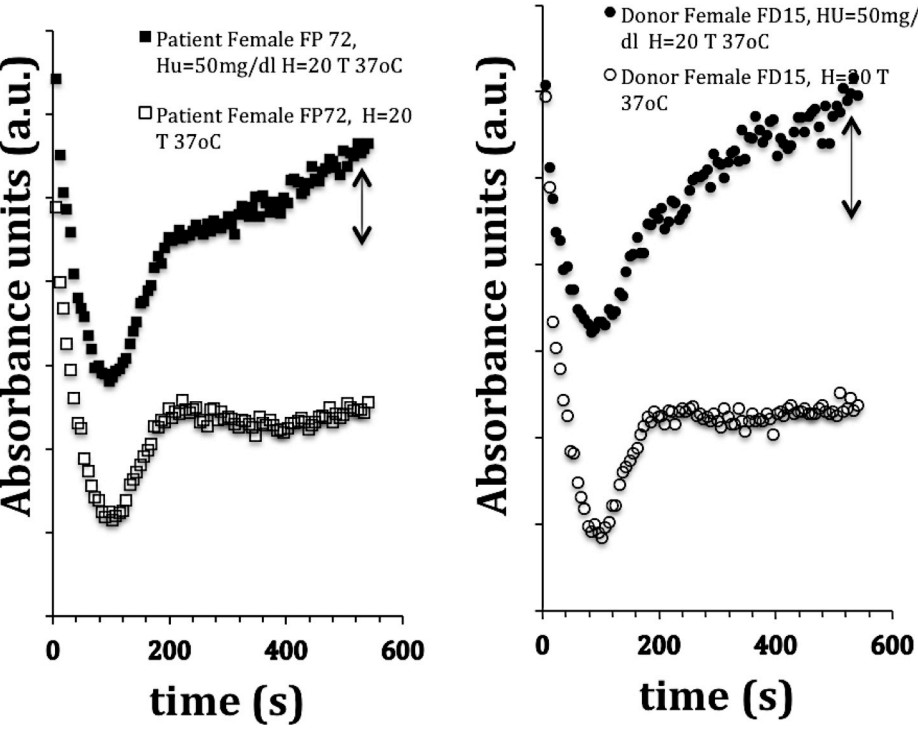

**Fig 7.** (a) and (b): Alterations of the Rigidity Index (Ri) of female patient (a) and a female donor (b) blood sample (Hct = 20, essay temperature: 37˚C) with the in vitro addition of hydroxyurea (HU) at a concentration of 50mg /ml. Ri value for patient FP72 without HU: 8.6; RI value with HU: 25.9 (p <0.001); Ri value for donor FD15 without HU: 5.7; RI value with HU: 166.7 (p <0.001). Values are the mean of three essays of the female patient FP72 and two essays for the female donor FD15 both with and without addition of HU, hematocrit adjusted for Hct = 20(0,2l/l), patient age: 13 y.o and Hb: 6.8 g/l; donor age: 47, Hb: 13.1 g /l, Test temperature: 37˚C.

hand, the correlations between TAMMV and Hb, Hct are negative, also observed in other studies [87]. The positive correlations of TAMMV with RDW and Ri indicate that RDW and the index of rigidity Ri have more direct physical nexus with the measures of velocity by TCD than Hct and Hb, especially since they relate to stiffness of the red blood cell and to its dispersion of sizes (RDW), which alter blood flow in the arteries. The two parameters RDW and Ri affect the blood flow in the cerebral arteries, but TAMMV and Ri have higher correlation values than those presented with RDW.

Fig 10 shows the correlations (Spearman's rho and Pearson's r) between the PI Gosling pulsatility index and the Ri rigidity index for the patients (see **S1 Table** at Supplemental Material for n and P-values presented in Fig 10). The correlations shown in Fig 10 between Ri and PI are significant and direct (stiffness grows, PI grows) but are weak. Only the correlation Ri vs. PI MCA-R has a moderate correlation (r = 0.56 p <0.001 n = 35). Physically, the higher the concentration of irreversibly stiffened cells, the higher the PI. The correlation between the Ri of the native blood sample (without Hct adjustment, only EDTA added) and the maximum PI value found in the TCD showed a significant correlation but weak r = 0.4 (p <0.02 n = 35).

Fig 11 shows these correlations (PI vs. Ri) obtained for 36 patient samples. Almost all are from SS patients (32 out of 36 patients). The curves obtained have shallow correlation values, as previously observed. However, they indicate a significant tendency that the increase of the Ri value occurs with the rise of the PI value. Moreover, this correlation does not seem to be linear. In the case of Hct = native, we correlate with the maximum of PI found in the TCD exam. It must say that -in searching and recording de Gosling PI data from the medical archives- we

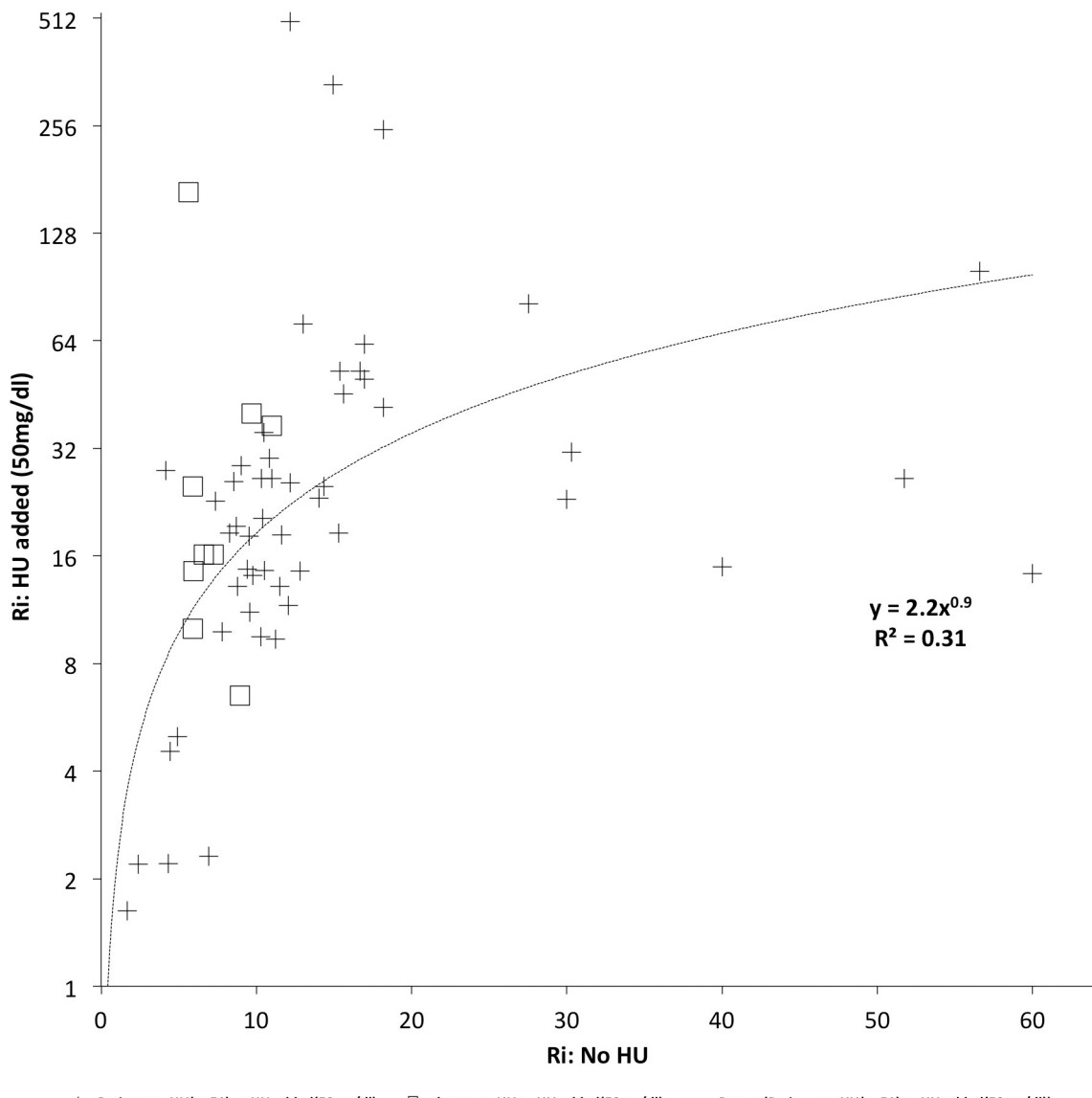

**Fig 8. All experiments with Hu added: relationship between Ri values for an aliquot of a blood sample from patients (n = 51) and donors (n = 8) who did not use either HU therapy (abscissa) and aliquot the same sample with the addition of 50 mg/ml.** HU Addition method described in item 2.3.6; test temperature 37 °C.

sought to approximate or coincide the day of the PI/TCD examination with the day of the test for determination of Ri.

No significant correlations observed between PI vs. Hb, PI vs. Hct, and Pi vs. RDW (n = 35). PI is considered a descriptor of distal resistance to cerebral blood flow (CBF) and the direct relationship between PI and Ri together with the lack of significant correlation between PI and Hct seem to indicate that mechanical resistance to CBF could be more correlated with the rigidity of the RBC and less with their volumetric fraction (Hct). Hct is a determinant of the viscosity of blood. On the other hand, the correlations between the PI left and PI right (rho = 0.83 p <0.001) are highly significant.

**Spearman's rho and Pearson's r correlation coefficients**

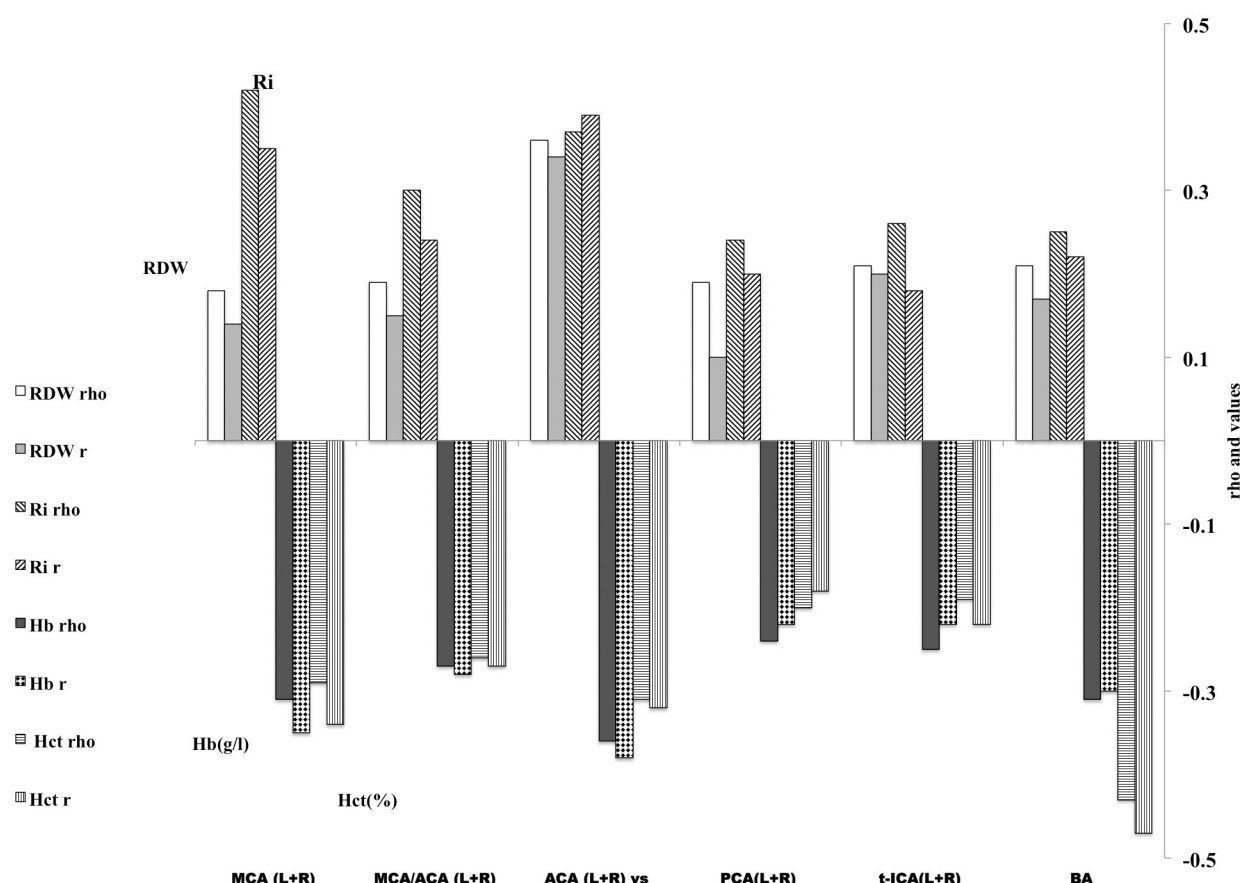

**Fig 9. Spearman ρ and Pearson r correlation values between TCD velocities (TAMMV in cm/s) and rigidity index Ri, RDW, Hb, and Hct.** Cerebral arteries MCA, MCA / ACA, ACA, PCA, t-ICA, BA, left and right. Data of patients SCA, SCC, SD, and Sβ0thal. Note that correlations are <u>direct</u> (positive values) between TCD velocities vs. (versus) Ri, and TCD vs. RDW, and inverse (negative values) between TCD velocities vs. Hb, and vs. Hct. Ri measured at Hct = 20 (0.2l/l); TAMMV-TCD, RDW, Hb, and Hct from Hemominas medical archives.

## 3.3. Rigidity index Ri (at Hct = 20 and Hct = native), Hb, Hct, RDW for patients and healthy donors

Table 3 shows the correlations between the Ri indexes with the hemoglobin concentration Hb (g /l), Hct (%), and RDW. The RBC rigidity Ri of patient samples is significantly and negatively correlated (Spearman Rho = -0.36, p <0.001 and Pearson r = -032, p <0.001) with Hb. Although the association is weak, the two parameters are associated, and all correlations calculated are inverses, i.e., the RBC rigidity index Ri increases with the reduction of Hb concentration. For blood from donors (Hct = 20), no association observed between the rigidity parameters Ri and the concentration of Hb. For the native blood sample, the inverse correlation between Ri and Hb is significant (Pearson r = -043; p <0.002), indicating that the overall rigidity index Ri of the red blood cells is strongly affected by the amount of irreversible RBC in the sample.

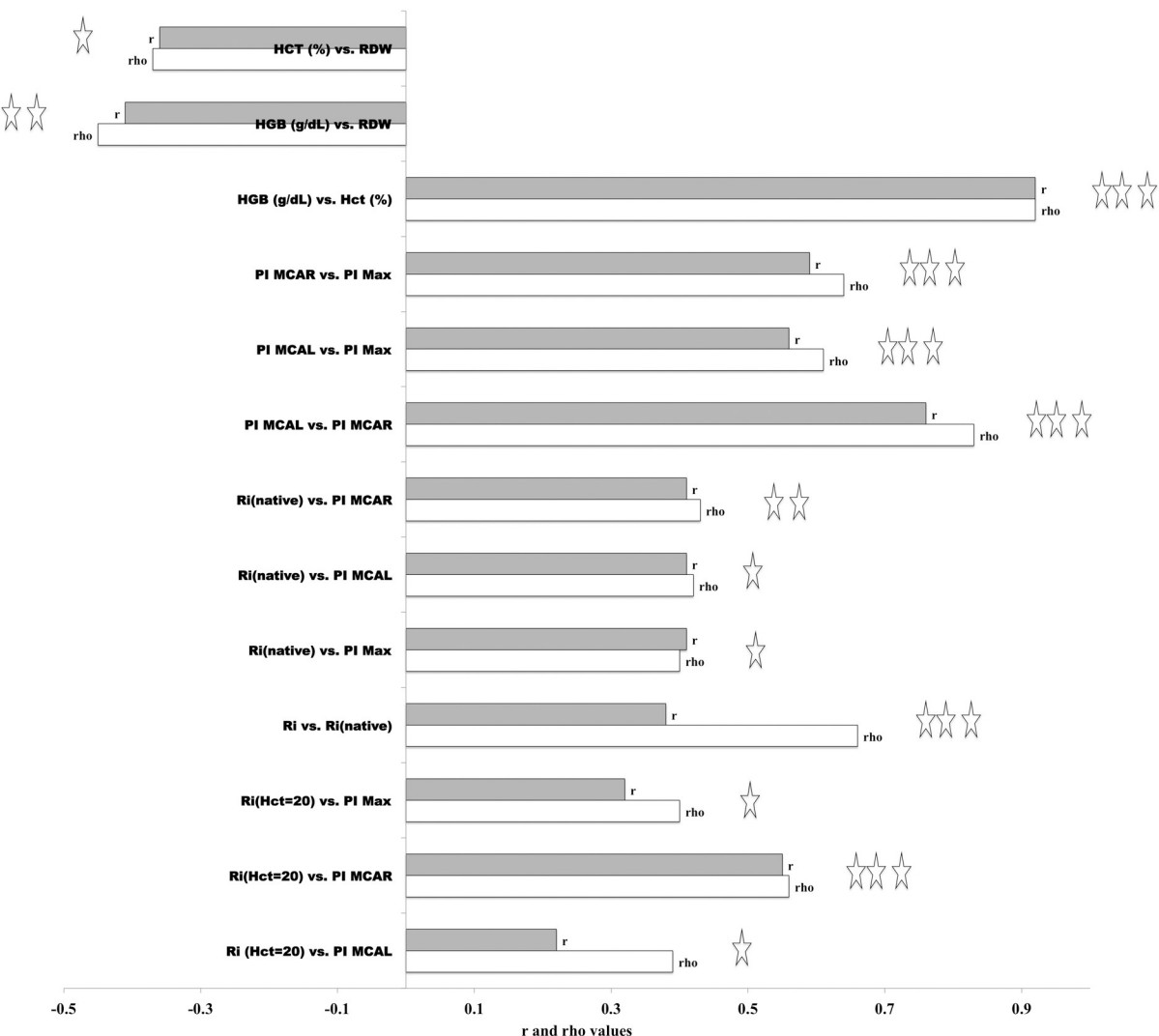

**Fig 10. Spearman (Rho) and Pearson (r) correlations between PI Gosling pulsatility index and Ri rigidity index for patients SCA, SCC, SD, Sbeta0thal.** It also showed correlations between PI vs. Hb, Hct, and RDW. Patients: SS: 32, SC: 2, Sbeta0thal: 1, SD: 1.

The correlation between the rigidity index Ri and RDW is positive, indicating that increased rigidity of the red blood cell is associated with increased RDW.

On the other hand, the relationship between RDW and Hb, RDW, and Hct is inversely indicating that the lower these values, the higher the variability in the red blood cell size distribution (RDW). This inverse correlation observed [88] since the introduction of automatic cell counters more than four decades ago. For these latter correlations, the values of n are higher (n> 300) because we use data from hemograms made at different dates for the same patient.

## Discussion

### 4.1- Characteristics of the technical principle of RBC rigidity measurements Ri

Can changes in the absorbance of a blood sample, during the sedimentation process, give information (semi-quantitatively) about the stiffness of these red blood cells? Yes, when there is a population of red blood cells in the sample that irreversibly polymerizes, that is, even when

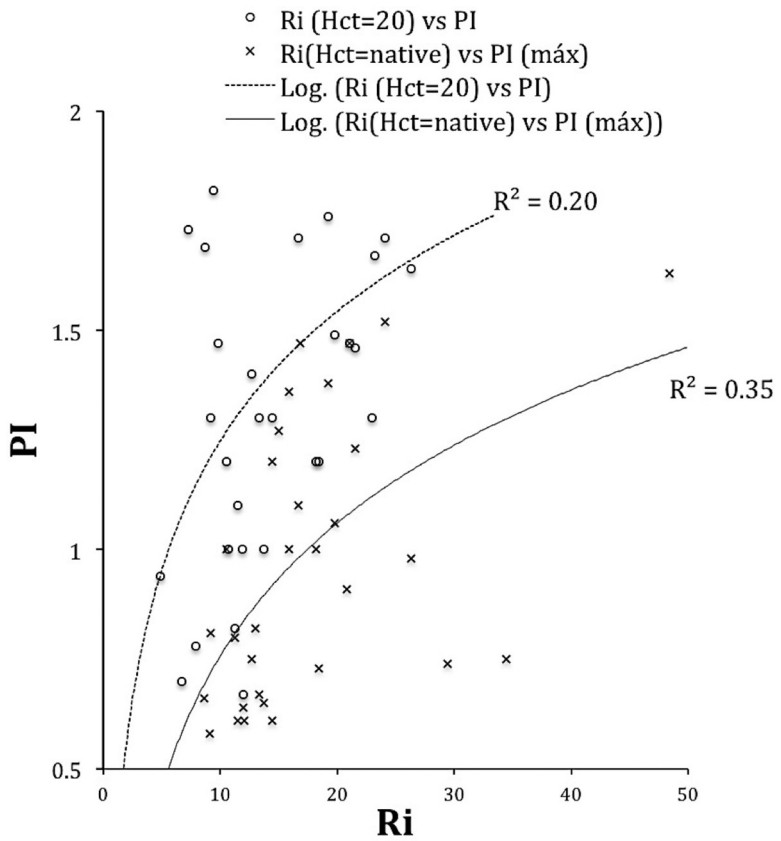

**Fig 11. Relationship between values of Ri measured with Hct = 20 (n = 36) and Ri measured with Hct = native.** Patients (M&F): SS: 32, SC: 2; Sbeta0thal: 1, SD: 1, R2 values shown in graph; Ri measurements at 37 $^{o}$C. Adjustment performed with power curve for Ri (Hct = 20).

oxygenated, no depolymerization occurs (or only partially). These stiffened red cells do not aggregate, as a central point for aggregation to happen is the deformability of the red cells. The difference between the initial and final absorbance values at 37 $^{o}$C indirectly and roughly indicates the concentration of this type of red blood cells in the sample. The smaller the difference between the initial and final values, the higher the frequency of polymerized red cells or any

**Table 3. Correlations between Ri(rigidity index) and RDW, Hb, and Hct: Spearman rho and Pearson r.**

| | Patients(SCA,SCC,Sβ0thal,SCD) | | | | | | | | | Healthy Donors (M&F) | | |
|---|---|---|---|---|---|---|---|---|---|---|---|---|
| | RDW | | | RI (Hct = native) | | | RI (Hct = 20) | | | RI (Hct = 20) | | |
| | n | ρ | r | n | ρ | r | n | ρ | r | n | ρ | r |
| Hb (g/dL) | 305 | -0.56*** | -0.55*** | 58 | -0.40*** | -0.43*** | 95 | -0.36*** | -0.32*** | 18 | -0.13$^{ns}$ | -0.001$^{ns}$ |
| Hct (%) | 302 | -0.46*** | -0.46*** | 58 | -0.39*** | -0.41*** | 77 | -0.36** | -0.35*** | - | - | - |
| RDW | | 1 | 1 | 94 | 0.25** | 0.21** | 95 | 0.37*** | 0.31*** | - | - | - |
| RI (Hct = 20) | 94 | 0.25** | 0.21** | | - | - | | 1 | 1 | 1 | 1 | 1 |

ns: not significant

* $p < 0.05$

** $p < 0.01$; p<0.001

***; RDW,Hb, and Hct from Hemominas medical archives.

other condition that prevents adhesion between them (red blood cell area/volume ratio, red cell surface changes, etc.).

One can also argue that several blood properties could affect the measurement of Ri, such as viscosity, sedimentation rate, fibrinogen concentration, and electrical charge on the surface of red blood cells (Zeta potential). Fibrinogen concentration and viscosity affect the sedimentation rate, but as we work with initial and final absorbance values, what occurs during the process does not change the difference between the initial and final absorbance values. The fact is that at Hct = 20 or Hct = native after about 200 seconds, the sample has ultimately settled to both donors and patients. Another 340 seconds are provided to verify that the sedimentation is stable.

In vitro addition of HU (T = 37˚C) produced massive changes in the Ri index values, as shown in Figs 6,7 and 8. It means that after sedimentation, even donor red cells did not remain adhered to each other. In Fig 7, after sample sedimentation (~ 200 s), the absorbance of the donor sample added with HU rather than stabilizing continues to increase, indicating that red blood cells are almost spontaneously separating from the rouleaux. This phenomenon is only possible if the effect of electric charge (Zeta potential) on the surface is higher than the adhesion forces between the red cells. As already noted, the wavelength used (655 nm) is outside the hemoglobin absorbance range. This HU effect also depends on the assay temperature because micrographs of in vitro HU-added samples obtained at room temperature show rouleaux. See S2 and S3 Figs HU-added Microstructure of blood samples obtained by light microscopy at Supplemental Material.

At a temperature of T = 37 °C, the Yield Shear Stress (in mPa) of blood is almost half the value of T = 25 °C (see ref. [10], page 150), and the rouleaux very quickly are undone at this temperature (T = 37 °C). So when formed, they can be so unstable that they tend to spontaneously separate. At body temperature, RBC is a liquid within another liquid [3]. From Fig 8, it can be deduced that the in vitro addition of HU immediately affects the electrical charge on the surface of red blood cells (zeta potential) since HU is an electrolyte. Recent experiments demonstrate the ability to change the zeta potential of particles by the addition of HU [89]. This effect could not be observed by light microscopy at room temperature (~ 25 oC), as shown in S2 and S3 Figs HU-added microstructure of blood samples obtained by light microscopy at Supplemental Material. Room temperature and surface tension of blood film between slides should exceed the effects of electrical charge observed at 37 °C.

Therefore the very high values of Ri obtained with in vitro addition of HU are not due to rigidification. With the addition of HU at 37 °C, the final absorbance is very close to the initial value, that is, the rouleaux breakaway is facilitated. An immediate effect of HU is the reduction of the Yield Shear Stress value required for the collapse of rouleaux. Future studies with the present technique will investigate the Ri in pathologies characterized by hyper aggregation (diabetes, hypertension) to evaluate the condition of RBC rigidity and hyper aggregation. However, even if hyper aggregation and stiffness simultaneously occur, the main effect on the arterial circulation is the presence of rigid RBCs altering the flow, which may affect the function of leukocytes. [90]. Recent studies discuss the effect of rigid cells on margination (movement towards endothelial walls) [91], and results, unfortunately, are not conclusive [92].

The technical principle proposed here has antecedents in the erythrocyte sedimentation rate (ESR). The main feature of the rigidity of SS erythrocytes, especially irreversible sickle cells, is that they do not form the usually very long (and flexible) "piles of coins" characteristic of healthy deformable RBCs called rouleaux, which decreases the absorbance of the sample. The higher the number of stiffened and isolated red blood cells, the higher the absorbance values. Our technical principle, however, uses to its advantage the major deficiencies of ESR, which is strongly influenced by hematocrit, erythrocyte aggregation rate, and plasma content.

A sample with Hct = 40 has abs40> abs20> abs10, etc. However, the difference [Ainitial-Afinal] is related to the properties of red blood cells, their ability to aggregate or to remain separate. The higher the aggregation (for whatever reason), the greater this difference, as the total area of the red blood cells is reduced. For situations of erythrocyte hyper-aggregation (diabetes, hypertension), specific future studies will be required. Absorbance measured at lambda = 655nm is very close to the value used by commercial ecktacytometer [93], which is 670 nm. At 655 nm wavelength, the effects of absorbance due to the presence of free Hb are minimized. The shear stress obtained with a micropipette (30 Pa) is sufficient to separate the red blood cells, and shear rates greater than 1000 s$^{-1}$ achieved, as indicated in 2.3.6.

Some techniques individually measure red cell deformability by imposing shear stress on individual cells, such as micropipette aspiration, optical tweezers, deformation within a microchannel, etc. In another technique—which includes the different commercial types of ecktacytometer [94]—shear stress is exerted on a blood sample, and deformability is a value derived from the sum of the contributions of diffracted RBCs that were in the path of a monochromatic light beam. It is an average value of deformability. Since the introduction of ecktacytometry to measure deformability, the difficulty in evaluating rigid red cells [95] such as SS red cells and spheroid red cells [96] with this technique was observed. Recent articles indicate that these deficiencies could be remedied in a particular trademark of ecktacytometer [97,98,99].

The fact that the method is static, of sedimentation, does not detract, especially considering that both this and ecktacytometry measure mean values of rigidity/deformability and not the rigidity/deformability of individual red blood cells. [100]. That is why call it the Ri overall rigidity index. Increasing or decreasing plasma viscosity also does not affect the Ri measurement, because what is measured is an optical property that is not influenced by a mechanical property. Furthermore, the data in Fig 3 indicate that differences in absorbance can discriminate between RBC rigidity of SCA patients treated with HU (n = 32) from the value obtained for donors (n = 40) and p <0.001. The effect of the combination of transfusion and HU therapies indicated that there is no significant difference between the Ri indexes of these patients and from healthy donors.

## 4.2 –Correlations between Ri and TCD (TAMMV and PI)

The most significant correlations obtained in this work are between the rigidity index Ri measured in our laboratory and the TCD data. Physically, the relationship between TCD velocities and Ri correlates with the fraction of stiffened red blood cells present in the sample. The higher the Ri, the more substantial the fraction of irreversibly deformed and non-aggregating sickle cells. The more significant this fraction, the higher the TAMMV values in all arteries probed by the TCD. Correlation values are significant but weak. The highest values of Spearman rho (0.42 ***, n = 80) and Pearson r (0.35 ***) occur in MCA L and R and indicate a tendency for the increase of Ri to accompany the rise of TAMMV. The correlations between RDW and TAMMV are also significant and direct, but they are weak correlations. The positive association between RDW and Ri with TCD velocities indicates that variability in red blood cell size and its fraction of rigid RBC is associated with higher speeds in brain flow.

More eloquent is the significant and direct correlation observed between PI and Ri obtained for MCA, the better correlation being with MCA-R (rho = 0.56, r = 0.55, p <0.001) in Hct = 20. Significant correlation values were also obtained between PI and Ri for MCA for Hct = native blood samples (rho = 0.43, r = 0.41, p <0.001). PI is considered an indicator of resistance to brain flow. Its direct relationship with Ri indicates that an increase in the fraction of stiffened RBCs is associated with an increase in the resistance to blood flow. As shown in **S1**

Table at Supplemental Material, no significant correlation observed between Pi and RDW, Pi and Hb, and Pi and Hct.

## Conclusions

Doctors and hematologists routinely express mention of blood viscosity and its importance to human health. Viscosity measurements, however, are not part of routine medical diagnostics. On the other hand, there is a growing understanding that red blood cell rigidity is the origin of hemoglobinopathies such as SCA and possibly other pathologies. The rigidity comes mainly from the gelatinization of HbS within the red cells. A fraction of these red cells become permanently rigidified in the circulatory system, and their harmful effects have not yet fully elucidated. TCD technology and blood count provide information that, if associated with a measurement of erythrocyte stiffness (or its inverse deformability), can describe a hemorheological picture of the patient without the need for viscosity measurements. In the present work, a curve similar to the sylectogram was used to investigate the rigidity of RBCs. The difference between the initial (6s) and final values (540s) of blood sample absorbance, for the same wavelength (655 nm) and hematocrit, is related to the fraction of red blood cells that did not aggregate due to being rigidified or to other changes yet to be elucidated. No substance other than EDTA anticoagulant needs to add to the sample, and only 100 ul of blood needed. There is no conclusive answer if the hematocrit affects Ri value in patients. However, considering that the volume of the sample is fixed (100ul), the concentration of irreversibly polymerized SS cells increases with Hct, and might affects the values of Ri, as shown for Hct = 20 and Hct = 40 of SCA patients. In the case of healthy donors, few experiments did not give a clear and significative answer, and more tests are needed. For the same hematocrit (Hct = 20), significant differences were observed between the Ri rigidity index values of SCA (and SCC) patients and the Ri values of healthy donors. SCA patients on transfusion or HU therapy have lower values than those without either therapy. With both therapies, patient Ri values do not differ significantly from donor Ri values.

The *in vitro* addition of HU alters both healthy and SS red blood cells so that Ri values for both categories are very high and close to each other in values. It is hypothesized, and to be investigated in future studies, that one of the main actions of HU, added *in vitro* at 37 $^{\circ}$C, would be to function as an electrolyte increasing the zeta potential of RBC, bringing it closer to values that facilitate rouleaux breakdown and separation at 37 $^{\circ}$C.

The most significant correlations obtained in this work are between the rigidity index Ri measured in our laboratory and the TCD data (TAMMV and Gosling PI). Physically, the relationship between the velocities measured by TCD (TAMMV) and Ri correlates with the fraction of irreversibly stiffened red blood cells present in the sample. We conclude that the higher the Ri, the more significant the fraction of irreversibly deformed and non-aggregating red blood cells. For some reason, the more significant this fraction, the higher the TAMMV values in all arteries probed by the TCD. It should be pointed out that the direct relationship between PI and Ri together with the lack of significant correlation between PI and Hct seem to indicate that mechanical resistance in the cerebral arteries is associated more directly with the rigidity of the RBCs and less with their volumetric fraction (Hct), which is determinant of the viscosity of blood. Finally, as observed by several authors and confirmed once again, the higher the Hct and Hb concentration, the lower the TAMMV values. The more intense the anemia, the higher is TAMMV.

The developed technical principle does not replace existing equipment, and this is not the purpose of the work. It also does not preclude future studies in pathologies that combine rigid red blood cells with hyper aggregation. The need to develop low-cost techniques to evaluate

red cell rigidity is a present issue for Hematology because the techniques used are restricted to the countries or regions of the world where they appeared. Throughout Brazil, with the sixth world population, we have news of one single piece of equipment [101]. Moreover, new experimental techniques to measure red cell deformability continuously appear, indicating the inexistence of a universalized method. It should be considered that blood centers in middle-income countries, for financial reasons, could only adhere to new techniques when the benefit introduced by the technology is very significant, as was the case in Brazil, of the use of TCD in children with sickle cell disease.

## Supporting information

**S1 Dataset. Patient and donor data: TCD-TAMMV, Gosling Pulsatility Index PI, Hct, Hb, RDW from Hemominas Foundation medical archives; absorbance measurements (96 points, 6–540 s) for each patient and Ri (rigidity index) calculations.**
(XLSX)

**S2 Dataset. Example of assays to calculate Ri (rigidity index) of an SS patient without HU and transfusion therapies.** Experiments at Hct = 20, Hct = native, and HU added in vitro.
(XLSX)

**S3 Dataset. Experiments with bovine blood (non-aggregating and deformable RBC) with and without fibrinogen.**
(XLSX)

**S4 Dataset. Experiments with blood absorbance in microplate well from 30 °C—40 °C.**
(XLSX)

**S1 Fig. Transmittance/absorbance dynamics of blood rouleaux formation inside the microwell.**
(TIFF)

**S2 Fig.** DONOR: No HU (top row) and HU-added (bottom row) microstructure of blood samples obtained by light microscopy.
(TIF)

**S3 Fig.** PATIENT: No HU (top row) and HU-added (bottom row) microstructure of blood samples obtained by light microscopy.
(TIF)

**S4 Fig. Contact angle measurements of commercial microplates.**
(TIF)

**S1 Sedimentation. Images from the side of a microplate well showing the process of sedimentation at the Hct = 20 sample from a donor.**
(ZIP)

**S1 Table.** (a) Spearman ρ and Pearson r correlation values between TCD velocities (TAMMV in cm/s) and Rigidity index Ri, RDW, Hb, and Hct. Cerebral arteries MCA, MCA / ACA, ACA, PCA, t-ICA, BA, left and right. Data of patients SCA, SCC, SD, Sβ0thal. (b) Rho and r correlations between PI Gosling Pulsatility Index and Ri Rigidity index for patients SCA, SCC, SD, Sbeta0 thal. It also showed correlations between PI vs. Hb, Hct, and RDW. Patients: SS: 32, SC: 2, Sbeta0thal: 1, SD: 1.
(XLSX)

## Acknowledgments

Our thanks go to the Patients, their Parents, and Donors for their kindness and to accept to participate in our research, to Hemominas staff especially Dr. Célia Maria Silva, MD, and Dr. André Belisário; to REDEMAT UFOP-UEMG for the institutional support and to Gabriel Leon de Queiroz Sousa on 1-collecting the samples and part of data at the medical archives at Hemominas Foundation and 2-painstakingly performing the experiments with the Microplate Reader; and to Prof. Maria Lúcia Malard for the help in reading and correcting the text.

## Author Contributions

**Conceptualization:** Antonio Valadão Cardoso.

**Formal analysis:** Antonio Valadão Cardoso.

**Investigation:** Antonio Valadão Cardoso.

**Methodology:** Antonio Valadão Cardoso.

**Project administration:** Antonio Valadão Cardoso.

**Validation:** Antonio Valadão Cardoso.

**Visualization:** Antonio Valadão Cardoso.

**Writing – original draft:** Antonio Valadão Cardoso.

**Writing – review & editing:** Antonio Valadão Cardoso.

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
