## [Decision Letter · Decision Letter 0]

4 Dec 2019

PONE-D-19-28834

An experimental erythrocyte rigidity index (Ri) and its correlations with Transcranial Doppler velocities (TAMMV), Gosling Pulsatility Index PI, hematocrit, hemoglobin concentration and red cell distribution width (RDW)

PLOS ONE

Dear Professor Valadão Cardoso,

Thank you for submitting your manuscript to PLOS ONE. After careful consideration, we feel that it has merit but does not fully meet PLOS ONE’s publication criteria as it currently stands. Therefore, we invite you to submit a revised version of the manuscript that addresses the points raised during the review process.

Normally I require more than one review; however, based on quality of the single review, and the desire of promptness, I made my decision on one review.  Please address all raised questions; also, I believe the reviewer meant, if you are NOT including EM pictures, do not describe protocol in Materials and Methods.

We would appreciate receiving your revised manuscript by Jan 18 2020 11:59PM. To enhance the reproducibility of your results, we recommend that if applicable you deposit your laboratory protocols in protocols.io, where a protocol can be assigned its own identifier (DOI) such that it can be cited independently in the future. For instructions see: http://journals.plos.org/plosone/s/submission-guidelines#loc-laboratory-protocols

We look forward to receiving your revised manuscript.

Kind regards,

Jeffrey Chalmers, Ph.D.

Academic Editor

PLOS ONE

Journal Requirements:

To CEMIG S / A for the financial support and the Undergraduate Grant to Gabriel Leon de Queiroz Sousa

Reviewers' comments:

Reviewer's Responses to Questions

**Comments to the Author**

1. Is the manuscript technically sound, and do the data support the conclusions?

Reviewer #1: Partly

2. Has the statistical analysis been performed appropriately and rigorously? 

Reviewer #1: Yes

3. Have the authors made all data underlying the findings in their manuscript fully available?

Reviewer #1: Yes

4. Is the manuscript presented in an intelligible fashion and written in standard English?

Reviewer #1: No

5. Review Comments to the Author

Reviewer #1: This paper analyzes the Red Blood Cell (RBC) rigidity index from healthy donors and unhealthy young patients (2-17 years). The author employs a straightforward photometric technique to measure the rigidity of the samples. The main goal of this research was to verify if the overall rigidity of RBCs is higher in unhealthy (sicke) cells obtained from young donors. The samples were obtained from 71 donors and 98 patients. The technique employed to measure the rigidity index is simple and the required equipment is available in low-resource areas, which makes it very interesting. Furthermore, the author tried to correlate this value with other parameters (Hct, Hb, velocity, Gosling's pulsatility index and Red Cell Distribution Width) to improve understanding of flow in cerebral arteries. Nevertheless, a poor and inconclusive correlation between the Ri and the rest of the parameters was found. Moreover, the manuscript is long, it contains multiple grammar mistakes and some figures are not important and could be moved to the supplemental materials. For the previous reasons, I do not recommend its publication unless the following issues are addressed:

- The patients and blood donors have different range of ages (from 2-17 and 20-65, respectively). Why did the author compare blood samples from underage patients against adult donors? Can we expect any difference in the measured quantities based on the age of the donors?

- If electron microscopy images are going to be presented in future publications, the author should remove the explanation regarding the procedure for taking these images from materials and methods section. The same applies to the procedure for recording the sedimentation process by optical microscopy.

- Author stated that “The values of Hct, Hb, and RDW obtained from the archive files are from the date of sample collection for the test of the rigidity index Ri or the time closest to that day”. Regarding the values that were obtained in a different day, I recommend the author to include the time that had passed between the Ri measurement and the Hct, Hb or RDW measurements, and also, to take into account that Hct and Hb could have changed if several days had passed between measurements.

- In Figure 3, the author explained that the difference between the initial and final absorbance values for the donors is higher because the deformable red cells form rouleaux. What does it mean? Do the RBCs form aggregates and different structures when they are sickle? Also, in Figure 3, I wonder if the slope of the curve at times between 100-200 s could give more information, as it looks like the slope is more pronounced for donors in comparison to unhealthy cells.

- HbS (sickle) and HbF (fetal) hemoglobin should be explained and introduced before their abbreviations. Also, the use of acronyms and abbreviations of other terms should be checked. What is the meaning of SCC, Sβ0thal, SD, MCA, ACA, ICA, PCA, BA?

- In Figure 4, the difference in the Ri value for the SCA patient is attributed to the change in Hct, however, this value does not change with Hct for healthy donors. Since the Hct may vary from donor to donor and also at different times for the same donor, I wonder if the value of Hct should be fixed before running the experiments or should be kept constant at the native value for all the donors.

- Table 2.A and Table 5 are too wide and are not correctly visualized, and Tables 3 and 4 are very difficult to read, I believe these could be better understood if presented as a Figure.

- There are several grammatical mistakes in the manuscript. I suggest the author to thoroughly review the whole manuscript before submission.

6. PLOS authors have the option to publish the peer review history of their article (what does this mean?). If published, this will include your full peer review and any attached files.

Reviewer #1: No

---

## [Author Response · Author response to Decision Letter 0]

17 Jan 2020

Belo Horizonte January 16, 2020

PONE-D-19-28834

An experimental erythrocyte rigidity index (Ri) and its correlations with Transcranial Doppler velocities (TAMMV), Gosling Pulsatility Index PI, hematocrit, hemoglobin concentration and red cell distribution width (RDW) –

PLOS ONE

Dear Professor Jeffrey Chalmers,

I separated into items each of the criticisms made by you and the Reviewer and tried to answer them as best as possible below.

Grateful for your attention and 

Cordially,

Antonio Valadão Cardoso, Professor at Minas Gerais State University – Brazil

1- Questions raised by Academic Editor (in italics in the file "Response to Reviewers"):

1.1- "…I believe the reviewer meant, if you are NOT including EM pictures, do not describe protocol in Materials and Methods."

Answer: EM (Electron Microscopy) protocol mention removed from the manuscript.

1.2- "Please ensure that your manuscript meets PLOS ONE's style."

Answer: Manuscript meets PLOS ONE's style.

1.3- "Please removes any funding-related text from the manuscript." 

Answer: Funding-related text removed.

1.4- "And let us know how you would like to update your Funding Statement. Currently, your Funding Statement reads as follows: The funders had no role in study design, data collection, and analysis, decision to publish, or preparation of the manuscript."

Answer: Funding Statement maintained. 

1.5- "While revising your submission, please upload your figure files to the Preflight Analysis and Conversion Engine (PACE) digital diagnostic tool."

Answer: All figure files approved by PACE

2- Questions raised by Reviewer (in italics in the original file "Response to Reviewers"): 

2.1. "Moreover, the manuscript is long, it contains multiple grammar mistakes…" 

Answer: 496 words of the text were cut (~ 5% of the total). Additionally, two tables were cut, and two figures transferred to supporting material at the suggestion of the reviewer. 

The text has been completely revised, and the grammatical errors corrected. To check for grammatical errors, I paid online grammar correction services. Also, I asked a colleague (Ph.D. in the UK) to do a final text check and correct any remaining grammatical errors. 

2.2- "…and some figures are not important and could be moved to the supplemental materials."

Answer: I cut two figures (figures 2 and 10) and transferred them to the supplementary material section.

2.3- "The patients and blood donors have different range of ages (from 2-17 and 20-65, respectively). Why did the author compare blood samples from underage patients against adult donors? Can we expect any difference in the measured quantities based on the age of the donors?"

Answer: Sickle cell stiffness is the result of gelatinization (partial or total) of the hemoglobin suspension, containing mutant S hemoglobin. Hemoglobin S polymerizes into fibers when this S concentration reaches a particular value. The deformability of healthy red blood cells is mainly due to hemoglobin remaining in a stable suspension. The difference in age ranges, in this case, does not affect the outcome when the S concentration is high enough to initiate polymerization, regardless of age (adult or child). Also, Hemominas Foundation (acronym HBH, the blood center that provided the samples) only accepts blood from donors older than 18 years. 

2.4- "If electron microscopy images are going to be presented in future publications, the author should remove the explanation regarding the procedure for taking these images from materials and methods section."

Answer: Removed. 

2.5- The same applies to the procedure for recording the sedimentation process by optical microscopy.

Answer: Removed. 

2.6- "Table 2.A and Table 5 are too wide and are not correctly visualized…"

Answer: Tables have been reformatted, and visualization is correct now.

2.7- "… and Tables 3 and 4 are very difficult to read, I believe these could be better understood if presented as a Figure."

Answer: Tables 3 and 4 are, now, presented as Figures 9 and 10.

2.8- Author stated that "The values of Hct, Hb, and RDW obtained from the archive files are from the date of sample collection for the test of the rigidity index Ri or the time closest to that day". Regarding the values that were obtained in a different day, I recommend the author to include the time that had passed between the Ri measurement and the Hct, Hb or RDW measurements, and also, to take into account that Hct and Hb could have changed if several days had passed between measurements.

Answer: This part of sub-item 2.5.1 changed to: 

…………………………………………………………………………………………………

Data collection in the HBH files was performed directly in the patient's record. In possession of the date of blood collection to measure the Ri, we searched the patient's folder for the blood count performed on the date closest to the date of the project collection. HBH has not permitted reproduction by any means of these patient medical records. It only allowed copying the blood count data. Since errors can be made in the process of collecting and transcribing data manually, the coincidence of dates between Ri measurement and blood count is not guaranteed. On the other hand, as it is a blood donation procedure, it is ensured that the donor hemoglobin concentration data (g / l) are from the same collection date to measure Ri from that donor.

2.9- In Figure 3, the author explained that the difference between the initial and final absorbance values for the donors is higher because the deformable red cells form rouleaux. What does it mean? Do the RBCs form aggregates and different structures when they are sickle? 

Answer: Microstructure differences, between healthy and SCA patient blood, are easily identifiable under the optical microscope. To observe under the optical microscope, we placed a drop of blood, with EDTA anticoagulant only, on the glass slide and gently deposited a coverslip over this drop. Deformable donor red blood cells form very large, stable, and interconnected rouleaux when the blood is at rest. Like large stacks of coins, fitted precisely on top of each other. SCA RBC forms smaller, poorly structured, disconnected rouleaux. Irreversibly polymerized SS erythrocytes generally do not form rouleaux and are generally separate, individualized.

2.10- Also, in Figure 3, I wonder if the slope of the curve at times between 100-200 s could give more information, as it looks like the slope is more pronounced for donors in comparison to unhealthy cells.

Answer: This Reviewer remark has been added to the text (line number 323). In the case of donors, possibly between 100-200s, the already settled and connected rouleaux form 3-D structures and contract, increasing absorbance. In the case of deformable donor red blood cells, this contraction (increased absorbance) is faster because there is a 3D connection between the rouleaux. In the case of patients, the connection between rouleaux is much smaller, and the increase in absorbance occurs more slowly.

2.11- HbS (sickle) and HbF (fetal) hemoglobin should be explained and introduced before their abbreviations. 

Answer: HbS introduced and explained starting at line 61, and HbF at line 93.

2.12 - Also, the use of acronyms and abbreviations of other terms should be checked. What is the meaning of SCC, Sβ0thal, SD, MCA, ACA, ICA, PCA, BA?

Answer: All full name acronyms and abbreviations added to the text.

2.13- In Figure 4, the difference in the Ri value for the SCA patient is attributed to the change in Hct, however, this value does not change with Hct for healthy donors. Since the Hct may vary from donor to donor and also at different times for the same donor, I wonder if the value of Hct should be fixed before running the experiments or should be kept constant at the native value for all the donors.

Answer: This Reviewer remark has been added to the text (line number 368).

2.15- There are several grammatical mistakes in the manuscript. I suggest the author to thoroughly review the whole manuscript before submission.

Answer: A thorough revision of the entire manuscript was made. The grammatical errors found were corrected, as can be seen in the file "Revised manuscript with track changes."

---

## [Decision Letter · Decision Letter 1]

30 Jan 2020

An experimental erythrocyte rigidity index (Ri) and its correlations with Transcranial Doppler velocities (TAMMV), Gosling Pulsatility Index PI, hematocrit, hemoglobin concentration and red cell distribution width (RDW)

PONE-D-19-28834R1

Dear Dr. Valadão Cardoso,

We are pleased to inform you that your manuscript has been judged scientifically suitable for publication and will be formally accepted for publication once it complies with all outstanding technical requirements.

With kind regards,

Jeffrey Chalmers, Ph.D.

Academic Editor

PLOS ONE

Additional Editor Comments (optional):

Reviewers' comments:

Reviewer's Responses to Questions

**Comments to the Author**

1. If the authors have adequately addressed your comments raised in a previous round of review and you feel that this manuscript is now acceptable for publication, you may indicate that here to bypass the “Comments to the Author” section, enter your conflict of interest statement in the “Confidential to Editor” section, and submit your "Accept" recommendation.

Reviewer #1: All comments have been addressed

2. Is the manuscript technically sound, and do the data support the conclusions?

Reviewer #1: (No Response)

3. Has the statistical analysis been performed appropriately and rigorously? 

Reviewer #1: (No Response)

4. Have the authors made all data underlying the findings in their manuscript fully available?

Reviewer #1: (No Response)

5. Is the manuscript presented in an intelligible fashion and written in standard English?

Reviewer #1: (No Response)

6. Review Comments to the Author

Reviewer #1: (No Response)

7. PLOS authors have the option to publish the peer review history of their article (what does this mean?). If published, this will include your full peer review and any attached files.

Reviewer #1: No

---

## [Editor Report · Acceptance letter]

6 Feb 2020

PONE-D-19-28834R1 

An experimental erythrocyte rigidity index (Ri) and its correlations with Transcranial Doppler velocities (TAMMV), Gosling Pulsatility Index PI, hematocrit, hemoglobin concentration and red cell distribution width (RDW) 

Dear Dr. Valadão Cardoso:

I am pleased to inform you that your manuscript has been deemed suitable for publication in PLOS ONE. Congratulations! Your manuscript is now with our production department. 

With kind regards,

on behalf of

Dr. Jeffrey Chalmers 

Academic Editor

PLOS ONE